# KAAN: Kolmogorov-Arnold Activation Network — a Flexible Activation Enhanced KAN

## Abstract

Kolmogorov-Arnold Networks (KANs) have led to a significant breakthrough in the foundational structures of machine learning by applying the Kolmogorov-Arnold representation theorem. Through this approach, the target conditional distribution is expressed as the summation of multiple continuous univariate B-spline functions. The unique and complex computational structure of B-splines makes it hard to understand directly since the properties of each grid are not determined by its own parameters but are also influenced by the parameters of adjacent grids. Besides, it is challenging to trim and splice at components level under B-spline. To address this issue, we analyze the structural configurations of Multi-Layer Perceptrons (MLPs) and KANs, finding that MLP can be represented in a form conforming to Kolmogorov-Arnold representation Theorem (KAT). Therefore, we propose MLP style KAN framework Kolmogorov-Arnold Activation Network (KAAN), which is more straightforward, flexible and transferable. To verify the flexibility and transferability of our approach, we extend it to Convolutional Neural Network (CNN). Also, we demonstrate that parameter sharing is beneficial not only for efficiency but also for effectiveness. KAAN shows better representation capacity than MLP on several benchmarks. Furthermore, our experiment results lead us to conclude that this method is feasible for integrating modern network approaches such as CNNs.

## 1 Introduction

Recently, the newly released foundational framework Kolmogorov-Arnold Network (KAN)(Liu et al. (2024b)), designed to replace Multi-Layer Perceptron(MLP), has garnered widespread discussion upon its release. KANs increase the expressive power and interpretability of the model by using learnable univariate functions on the edges instead of fixed node activation functions. They achieve this by parameterizing the activation functions with spline functions, thereby replacing the linear weight matrices in MLPs. Compared to an earlier line of research, called Trainable Activation Networks (TANs) or Learnable Activation Networks (LANs) (Apicella et al. (2021)) which provide a unified trainable activation function for each layer, KAN not only moves the activation operation to the edges of the neural network but also assigns a unique, independently parameterized activation function to each edge. LAN involves assigning parameters to traditional activation functions, using parameterized functions, or even replacing activation functions with neural networks, whereas KAN provide each edge with independently trained B-spline as its activation.

Both KANs and LANs encounter challenges when compared to traditional MLPs. The additional parameter dimensions introduced by LAN and KAN increase training difficulty, computational complexity, and risk of overfitting, especially seriously when every activation of KAN is independent. Since the unique computational approach of B-spline, in which outputs of B-spline are calculated recursively relying on control points, empowers B-spline with coherence, any adjustment to an internal component affects grids nearby, complicating when adding or pruning components. Aside from this inconvenience, this coherence and complex computation approach hinders humans from understanding the properties of the activation relying solely on parameters.

To improve the straightforwardness and flexibility, we delve into the structural configurations of MLPs and KANs, examining their correlation and underscoring the structural advantages of KANs. By means of formal transformations, we ascertain that MLPs also comply with the Kolmogorov-

Arnold Theorem (KAT), essentially functioning as one kind of KAN in both form and essence. Building upon this transformation, we introduce the Kolmogorov-Arnold Activation Network (KAAN), which is linear combination of activation functions. This network provides improved representation capacity and a more straightforward, flexible, and transferable organization of activation functions. In contemporary network architectures such as Convolutional Neural Networks (CNNs) or Recurrent Neural Networks (RNNs), the topology of the computation graph often deviates from that of the neurons, with the number of edges in the computation graph typically exceeding the number of neuron edges. We contend that in KAN-type networks, edges should be related to neuron edges rather than the edge of computation graph. Then, we extend KAAN into the field of CNN utilizing this contender and the transferable nature of KAAN. Moreover, we endeavor to illustrate the feasibility of integrating the KAN framework into CNNs.

The main contribution of this paper are summarized as follows. The first is to demonstrate that MLP is a kind of generalized KAN. The second is to introduce KAAN, an straightforward, flexible and transferable framework of KAN series. The last is to demonstrate the benefits for effectiveness brought by parameter sharing and extend KAAN into the field of CNN.

The organization of the rest of this paper is as follows. In Section 2, we delve into recent research on KANs and closely related LANs. In Section 3, we establish that MLPs can be viewed as a special case of KANs in a broader context. In Section 4, we introduce KAAN and a convolutional application of it. In Section 5, we provide experimental evidence to support the findings in Sections 3 and 4. Finally, in Section 6, we discuss the properties of our approach and potential future avenues for research.

## 2 RELATED WORKS

Since the introduction of KAN, many studies have been conducted based on the KAN framework. The most popular research area focuses on the application of this new structure to various problems. Most studies follow the idea of KAN, utilizing its excellent representation and fitting capacities to explore applications in physical (Peng et al. (2024), Kundu et al. (2024), Howard et al. (2024)), diagnostic (Yang et al. (2024)), human behavior study (Liu et al. (2024a)) or to address problems in graph neural networks (De Carlo et al. (2024), Kiamari et al. (2024), Bresson et al. (2024)). Other studies replace some components of traditional CNNs (Cheon (2024), Li et al. (2024), Bodner et al. (2024)) or RNNs (Xu et al. (2024b), Vaca-Rubio et al. (2024), Genet & Inzirillo (2024), Herbozo Contreras et al. (2024)) with KAN layers to handle computer vision or time series problems.They focus on how to use KAN, but do not address the computational and deployment difficulties associated with it.

LAN is a line of research very similar to KAN. LAN replaces the activation functions in MLPs with new complex parameterized functions (Yuen et al. (2021), Pratama & Kang (2021), Subramanian et al. (2024), Bodyanskiy & Kostiuk (2023)), parameterizing commonly used activation functions (Apicella et al. (2019), Bingham & Miikkulainen (2022)) or even neural networks (Zhang et al. (2022)). In LAN, commonly used learnable activation functions can be roughly divided into polynomial activation functions (Chung et al. (2016), Goyal et al. (2019)), polynomial spline activation functions (Fakhoury et al. (2022), Ducotterd et al. (2024), Aziznejad & Unser (2019), Bohra et al. (2020)), exponential family functions (Machacuay & Quinde (2024)), radial functions (Vieira (2023), Machacuay & Quinde (2024)), periodic functions (Rußwurm et al. (2023)), and wavelet basis functions (De Silva et al. (2020)). These studies provide directions for the choice of activation functions.

Since KAT does not impose restrictions on the nature of the continuous univariate functions used in the model, any continuous univariate basis functions can be used. It is apparent that functions with infinite discontinuities in the exponential family cannot be used. Besides, higher-order polynomial activation functions have inherent limitations, being completely surpassed by B-splines. Hence, besides polynomial spline functions, radial functions (Aghaei (2024), Abueidda et al. (2024), Li (2024), Ta (2024)), Fourier functions (Xu et al. (2024a)), and some wavelet functions (Bozorgasl & Chen (2024), Azam & Akhtar (2024), Seydi (2024)) are suitable for constructing neural networks that comply with KAT constraints. Unfortunately, although the similarity between Universal Approximation Theorem(UAT) and KAT has been noted (Dhiman (2024)), the absence of a bridge between MLP and KAN result in the non-existence of a general MLP-style KAN framework.

In the following sections of this paper, KANs will be used to refer to all neural networks constructed with layers that meet KAT constraints, rather than specifically referring to the standard case using B-splines.

# 3 MLPs ARE KANs

In this section, we demonstrate that MLP represents a specific instance of KAN.

## 3.1 DECOMPOSING COMPUTATION IN KAN AND MLP

An MLP can be described as the composition of multiple Single Layer Perceptrons (SLPs), while the $\ell$-th SLP with parameter $p^{(\ell)}$ can be described as the composition of a parameterized linear operator $\mathcal{L}^{(\ell)}(\cdot; p^{(\ell)})$ and a non-parameterized nonlinear activation $\sigma^{(\ell)}$. Consequently, an MLP network can be represented as:

$$\mathcal{F} = (\sigma^{(n)} \circ \mathcal{L}^{(n)}) \circ \cdots \circ (\sigma^{(2)} \circ \mathcal{L}^{(2)}) \circ (\sigma^{(1)} \circ \mathcal{L}^{(1)}) \tag{1}$$

Assume the input of the $\ell$-th layer is $x^{(\ell)} = (x_1^{(\ell)}, x_2^{(\ell)}, \ldots, x_i^{(\ell)}, \ldots, x_n^{(\ell)})$, then the output of the neuron $y^{(\ell)} = (y_1^{(\ell)}, y_2^{(\ell)}, \ldots, y_j^{(\ell)}, \ldots, y_m^{(\ell)})$ can be computed as in Equation 2 and shown in Figure 1a.

$$y_j^{(\ell)} = \sigma^{(\ell)} \left( \sum_{i=1}^n w_{ji}^{(\ell)} x_i^{(\ell)} + b_j^{(\ell)} \right), j = 1, \cdots m \tag{2}$$

Here, $w_{ji}^{(\ell)}$ represents the element in the $j$-th row and $i$-th column of the weight matrix $W^{(\ell)}$, and $b_j^{(\ell)}$ denotes the $j$-th element of the bias vector $b^{(\ell)}$. All of these weights are referred to as the edges of the neural network. Since each neuron first applies a linear transformation to the input and then passes the result through a nonlinear activation function, the activation function is typically considered to be located at the node rather than on the edge.

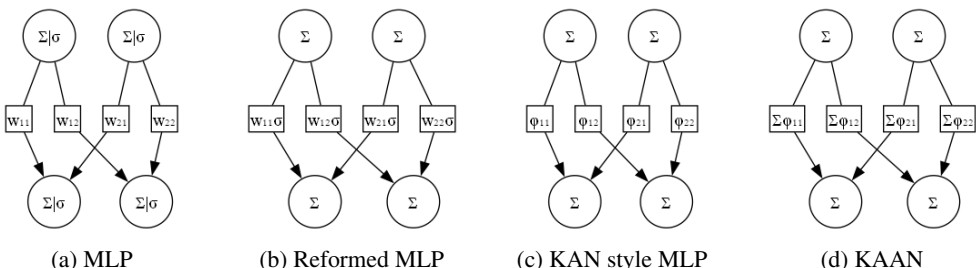

|        |                  |               |        |
|--------|------------------|---------------|--------|
| (a) MLP | (b) Reformed MLP | (c) KAN style MLP | (d) KAAN |

Figure 1: Different Structures of Nodes and Edges

KAN retains the topological structure of the MLP, while introduces significant modifications at a more detailed level. Firstly, KAN assigns trainable parameters to all activation functions, so each activation function no longer possesses fixed characteristics. This type of network, where a learnable activation function is shared across all neurons within each layer, is known as a LAN. Secondly, unlike in MLPs where activation functions are applied to the neurons of each layer, KAN applies the activation functions to the edges. In contrast to LANs, where the activation function parameters are shared across each layer, a distinctive feature of KAN is that the activation functions on each edge are independently trained, meaning each edge in a KAN has its unique activation function. The neurons in KAN sum the outputs of all the activation functions on the edges.

Since KAN requires its basis functions to be univariate continuous functions, let $\phi^{(\ell)}$ denote any parameterized univariate continuous function with parameters $p^{(\ell)}$, which is B-spline function in

standard KAN. Then, a generalized form of KAN neuron can be expressed as:

$$y_j^{(\ell)} = \sum_{i=1}^{n} \phi_{i,j}^{(\ell)}(x_i^{(\ell)}; p^{(\ell)}) \tag{3}$$

Next, we will demonstrate the uniformity in the forms of MLPs and this generalized KAN.

## 3.2 MLP is Generalized KAN

As in Equation 1, an MLP network comprises alternating parameterized linear transformations and non-parametric nonlinear activations. Since the composition of operators follows the associative property, we can rewrite Equation 1 into a new computational structure, which is equivalent in nature but different in form, as follows:

$$\mathcal{F} = \sigma^{(n)} \circ (\mathcal{L}^{(n)} \circ \sigma^{(n-1)}) \circ \cdots \circ (\mathcal{L}^{(2)} \circ \sigma^{(1)}) \circ \mathcal{L}^{(1)} \tag{4}$$

In this new structure, the neural network consists of three parts: a linear transformation preprocessing stage, several nonlinear processing layers composed of activation functions and linear transformations, and a post-processing layer with an activation function. Through this reassociation, the computation sequence of parameterized nonlinear transformations in the network changes from a linear-activation order to an activation-linear order. If we consider each node as merely performing a summation operation, then an activation-weighting operation is applied to the output features of the previous nodes, as shown in Figure 1b. As a result, the feedforward calculation of the $\ell$-th layer is as follows:

$$
\begin{aligned}
y_j^{(\ell)} &= \sum_{i=1}^{n} w_{ji}^{(\ell)} \sigma^{(\ell-1)}(x_i^{(\ell)}) + b_j^{(\ell)} \\
&= \sum_{i=1}^{n} \phi_{i,j}^{(\ell)}\left(x_i^{(\ell)}; w_{ji}^{(\ell)}\right) + b_j^{(\ell)}
\end{aligned}
\tag{5}
$$

By defining the composition of these two operations as a single activation function, i.e., $\phi_{i,j}^{(\ell)}(x; w_{ji}^{(\ell)}) = w_{ji}^{(\ell)} \sigma^{(\ell-1)}(x_i^{(\ell)})$, the operation on each edge transforms from a simple weighting function to a parameterized nonlinear activation operation as shown in Figure 1c. Under the premise that the activation function is restricted to a univariate continuous function (such as sigmoid or ReLU), this layer satisfies the requirements of KAT, thus forming a generalized KAN layer. Our recombination leverages the local structure $\sigma \circ \mathcal{L} \circ \sigma$ present in MLPs, where $\sigma$ satisfies the distributive and associative properties with respect to the multiplication of $\mathcal{L}$. When this structure was represented as $(\sigma^{(\ell)} \circ \mathcal{L}^{(\ell)}) \circ \sigma^{(\ell-1)}$ and $\sigma^{(\ell-1)}$ is treated as part of the previous layer, this structure forms an MLP layer. Conversely, when it is represented as $\sigma^{(\ell)} \circ (\mathcal{L}^{(\ell)} \circ \sigma^{(\ell-1)})$ and $\sigma^{(\ell)}$ is considered part of the next layer, this structure constitutes a KAN layer. Consequently, the MLP is a special case of a generalized KAN. Although this model may differ in practice from many modern network structures, such as CNNs, which do not follow the simple alternating pattern of linear and activation layers, the performance difference of networks with different sequences is minimal, as we will demonstrate in Section 5.1.

## 4 Approach

In this section, we introduce our network framework based on the idea that activation function could be any univariate continuous function and extend this approach to CNNs.

### 4.1 Kolmogorov-Arnold Activation Network

As mentioned above, both MLP and MLP-formed KAN consists of layers composed of two types of operations: activation and linear transformations. By reversing the arrangement typically used in MLP, each MLP layer is transformed into a KAN layer that adheres to the principles of KAT.

Furthermore, by replacing the standard Tanh activation function commonly used in MLP with an arbitrary parameterized function, each edge in the network can have a unique activation function. If the activation function is defined as a linear combination of multiple parameterized nonlinear functions, the definition of each edge becomes highly flexible and adaptable. When the neuron has multiple parallel basis functions, let $\phi_{i,j,t}$ represent the $t$-th component of the activation on the $i$-th input edge of the $j$-th neuron parameterized by $p_{i,j,t}$. As shown in Figure 1d, the feedforward function of the neuron is shown in the following equation:

$$
\begin{aligned}
y_j &= \sum_i \sum_t w_{i,j,t} \cdot \phi_{i,j,t}(x_i; p_{i,j,t}) \\
&= \sum_t \sum_i w_{i,j,t} \cdot \phi_{i,j,t}(x_i; p_{i,j,t}) \\
&= \sum_{i,t} w_{i,j,t} \cdot \phi_{i,j,t}(x_i; p_{i,j,t})
\end{aligned}
\tag{6}
$$

Similar to KAN, our framework assigns each edge an activation function. But instead of B-spline, each activation function is composed of a linear combination of multiple activation components and aggregates the outputs of these activations at the node. Therefore, we refer it as Kolmogorov-Arnold Activation Network (KAAN).

## 4.2 CONVOLUTIONAL KAAN

KAAN can not only be deployed in fully connected MLPs but can also be applied to modern neural networks, such as RNNs, CNNs, and Transformers, which heavily reuse neurons and exhibit differences in feature map computations and neuron topologies. We use CNN as an example to construct the Convolutional Kolmogorov-Arnold Activation Network (CKAAN).

When each activation exists on the edge of the neuron, for a convolutional layer with input $x \in \mathbb{R}^{H \times W \times C_\text{in}}$, output $y \in \mathbb{R}^{H' \times W' \times C_\text{out}}$, where $H$ and $H'$ represent the height, $W$ and $W'$ represent the width, and $C_\text{in}$ and $C_\text{ouy}$ represent the number of the channel respectively, the convolutional kernel can be descript as a $5$-dimensional tensor:

$$
f \in \mathbb{R}^{K_H \times K_W \times T \times C_\text{in} \times C_\text{out}}
\tag{7}
$$

where $K_H$ and $K_W$ represents the height and width of the convolution kernel respectively, and $T$ represent the number of activation components. The $t$-th component $f_{i,j,t,ic,oc}$ of the convolution kernel at position $(i, j)$, input channel $ic$, and output channel $oc$ is described by a weight parameter $w_{i,j,t,ic,oc}$ and a parameterizable activation function $\phi_{i,j,t,ic,oc}(\,\cdot\,; p_{i,j,t,ic,oc})$, where $p_{i,j,t,ic,oc}$ represents its parameters. Thus, the output of each neuron is computed by the following formula:

$$
y_{h',w',oc} = \sum_{i=0}^{K_H-1} \sum_{j=0}^{K_W-1} \sum_{ic=0}^{C_\text{in}-1} \sum_{t_a=0}^{T-1} f_{i,j,t,ic,oc}(x_{h'+i,w'+j,ic}; w_{i,j,t,ic,oc}, p_{i,j,t,ic,oc})
\tag{8}
$$

where

$$
f_{i,j,t,ic,oc}(\,\cdot\,; w_{i,j,t,ic,oc}, p_{i,j,t,ic,oc}) = w_{i,j,t,ic,oc} \cdot \phi(\,\cdot\,; p_{i,j,t,ic,oc})
\tag{9}
$$

Here, $y_{h',w',oc}$ represents the value of the output tensor at position $(h', w')$ for the $oc$-th channel, and $x_{h'+i,w'+j,ic}$ represents the value of the input tensor at position $(h'+i, w'+j)$ and channel $ic$.

Crucially, due to the different connection structures between feature maps and neurons, the independence of activation in modern network architectures may exist at either the feature map level or the neuron level. In CKAAN, the level of independence of the activation function is arranged at the neuron level rather than the feature map level. We validate the rationale for this arrangement in Section 5.1.

## 5 EXPERIMENTS

In Section 3.2, we claim that even if modern network does not conform to the structure of alternating linear and activation layers, changing the order of linear transformations and activations has little effect on the performance of the network. In Section 4.2, we set the independence of the activation function at the neuron level rather than at the feature map level. In this section, we verify these correctness of the statements in Section 5.1, and validate the performance of KAAN and CKAAN in Section 5.2.

In our experiment, apart from fixing the random seed for Toy dataset creation to the experiment ID ranging from 0 to 99 in Section 5.1, no random seed is fixed for any other random generators. All other aspects involving randomness, such as model parameter initialization and data input order, introduce stochasticity.

Standard Computer Vision(CV) datasets are used to demonstrate that our method can be extended to more complex network architectures beyond MLPs. Additionally, we incorporate a collection of toy datasets to evaluate the representational capacities of KAANs, as well as a collection of tabular task datasets specifically designed for KAN in Bench, to rigorously assess the performance of our approach.

**CV Datasets** For the CV tasks, we utilize four well-established datasets: MNIST (LeCun et al. (1998)), Fashion-MNIST (FMNIST)(Xiao et al. (2017)), CIFAR-10, and CIFAR-100 (Krizhevsky et al. (2009)). CIFAR-10 and CIFAR-100 are two prominent datasets frequently employed in image classification research, both developed by the Canadian Institute for Advanced Research (CIFAR). Similarly, MNIST and FMNIST serve as benchmark datasets for image classification tasks.

**Toy Datasets** SciPy (Virtanen et al. (2020)) offers several commonly used tools for generating synthetic datasets, and we select five of them: classification, moons, circles, blobs, and friedman1.

**Tabular Benchmarks** We use the collection of tabular benchmarks for KANs based on Poeta et al. (2024). This collection includes 8 tabular classification tasks, say Breast Cancer Wisconsin Diagnostic (BCWD)(Wolberg et al. (1993)), Spambase (Hopkins et al. (1999)), MAGIC Gamma Telescope (MAGIC)(Bock (2004)), Adult (Becker & Kohavi (1996)), CDC Diabetes Health Indicators (CDC), Dry Bean (dry (2020)), Statlog (Shuttle)(sta), and Poker Hand (Cattral & Oppacher (2002)).

In the following sections, we employ various potentially useful basis functions for fitting, each with distinct characteristics. Linear combinations of these basis functions can be used to construct activation functions. The Gaussian function, described by its mean $\mu$ and standard deviation $\sigma$, is a classic probability density function commonly used in statistics and signal processing. The Difference of Gaussians (DoG) function emphasizes edge features, often utilized in image processing and edge detection tasks. Fourier functions, represented as combinations of sine and cosine functions, are effective at capturing periodic features in data and widely used in signal and spectral analysis. Polynomial functions, are useful for modeling complex nonlinear relationships and play a crucial role in curve fitting, interpolation, and approximation problems. For the sake of training efficiency, we use only a subset of the parameters of these basis functions as trainable parameters, while the remaining parameters are predefined as hyperparameters . These basis functions are organized in Table 1, where superscripts denote given hyperparameters, while parameters shown as inputs to the functions represent the trainable parameters.

Table 1: Definitions of Basis Functions

| Basis Function | Definition |
|---|---|
| Gaussian | $f^{(\sigma)}(x; \mu) = \frac{1}{\sigma\sqrt{2\pi}} e^{-\frac{(x-\mu)^2}{2\sigma^2}}$ |
| DoG | $f^{(\sigma)}(x; \mu) = -\frac{(x-\mu)}{\sigma^3\sqrt{2\pi}} e^{-\frac{(x-\mu)^2}{2\sigma^2}}$ |
| Fourier | $f^{(n)}(x; a_n, b_n) = a_n \cos(nx) + b_n \sin(nx)$ |
| Polynomial | $f^{(n)}(x; a_n) = a_n x^n$ |

In order to increase the representation range, the means of the Gaussian and DoG functions are initialized to $\mu = -1, 0, 1$, and to maintain training stability, the variance is freezed $\sigma = 1$. We only select 4 lowest-frequency Fourier bases in order to avoid overfitting.

In this work, the polynomial activation functions include several configurations: a 4th-order polynomial, a linear combination of four 4th-order polynomials, and a 16th-order polynomial. The performance of KAAN with a 16th-order polynomial as the activation function is tested to highlight the unique drawbacks of high-order polynomials, as discussed in Section2.

Additionally, we design the ParallelV1 activation function as a linear combination of ReLU, SiLU, and Tanh activations. For comparative reference, we also consider the ParallelV2 activation function, which combines the ParallelV2 activation with the previously mentioned Gaussian, DoG, and Fourier activations.

Table 2: Activation Functions and Their Definitions

| Name | Definition | Abbreviation |
|---|---|---|
| Gaussian | Linear combination of Gaussian basis functions based on $\mu = -1, 0, 1$. | G |
| DoG | Linear combination of the derivatives of Gaussian functions. | DoG |
| Fourier | Using the 4 lowest-frequency Fourier basis functions. | F |
| Poly4 | Polynomial function with degree $n = 4$. | P4 |
| Poly4*4 | Linear combination of 4 polynomial functions of degree $n = 4$. | P4*4 |
| Poly16 | Polynomial function with degree $n = 16$. | P16 |
| ParallelV1 | Linear combination of ReLU, SiLU, and Tanh. | PV1 |
| ParallelV2 | Combination of Gaussian, DoG, Fourier, and ParallelV1. | PV2 |

In the following experiments, we denote the KAAN or CKAAN models based on the model name and the abbreviation of activation functions they utilize. For example, a KAAN with four degree polynomial activation is denoted as KAAN_P4, and a CKAAN with ParallelV1 activation is denoted as CKAAN_PV1.

## 5.1 VALIDATIONS

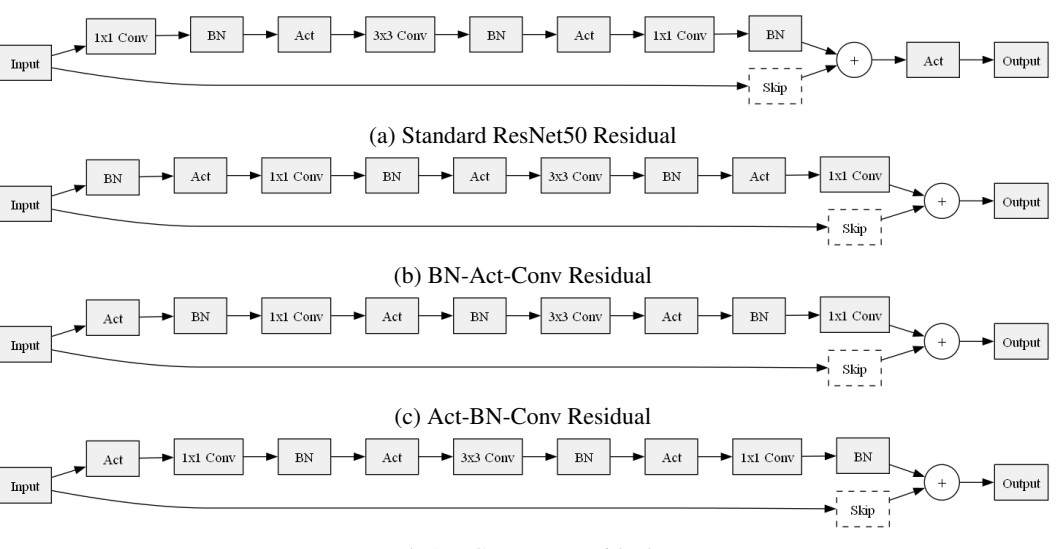

(a) Standard ResNet50 Residual

(b) BN-Act-Conv Residual

(c) Act-BN-Conv Residual

(d) Act-Conv-BN Residual

Figure 2: Re-arranged ResNet50 Residual Structures

In Section 3.2, we demonstrated through a simple form transformation that MLP also meets the requirements of KAT. However, for modern architectures that do not adhere to the simple alternating structure of linear layers and activation layers, the situation is not as straightforward. We claim that by directly swapping the order in which activation and linear layers appear to meet the requirements

of KAT, the performance will not be affected. And, we conduct a validation base on ResNet50 (He et al. (2016)). In the standard ResNet50 architecture, shown in Figure 2a, each residual unit follows the following sequence of operations: the first layer is a $1 \times 1$ convolution followed by Batch Normalization (BN) and ReLU activation; the second layer is a $3 \times 3$ convolution again followed by BN and ReLU activation; the third layer is another $1 \times 1$ convolution followed by BN, but without activation function, as the activation function is applied after the skip connection. We swap the order of CNN layers and activation layers to make it conform to the structure of the CKAAN, but the presence of the BN layers interfered with our experiment. Therefore, we design three different structures where activation comes before CNN, based on the position of BN layers: before activation as in Figure 2b, between activation and convolution in Figure 2c, and after convolution in Figure 2d. We test these four different structures on CV datasets.

As discussed in Section 4.2, the connection structures of feature maps and neurons are the same in fully connected architectures, but not in modern architectures. Therefore, we add a set of experiments to verify whether the independency of activation function is tied to the network topology of neurons or the computational graph structure. We implement a residual block based on the standard ResNet50 approach, where independence exists on feature maps.

Table 3: Accuracies of the standard ResNet-50, ResNet-50 with modified operation order, and the ResNet-50 without parameter sharing

| Dataset | Standard | BN-Act-Conv | Act-BN-Conv | Act-Conv-BN | Independent |
|---------|----------|-------------|-------------|-------------|-------------|
| **CIFAR100** | 67.43 | 67.06 | 66.93 | **68.54** | 43.03 |
| **CIFAR10** | 90.34 | 89.53 | **90.36** | 90.35 | 75.78 |
| **MNIST** | 99.38 | **99.50** | 99.42 | 99.49 | 99.22 |
| **F-MNIST** | 92.29 | **92.83** | 92.63 | 92.5 | 91.88 |

As shown in Table 3, when batch normalization is functioning effectively, changing the order of convolution and activation operations does not degrade the performance of the network. However, if the placement of batch normalization is suboptimal, applying activation before convolution may result in a certain degree of performance loss. Furthermore, abandoning the parameter sharing which is common in modern architectures would have a devastating impact on the model's performance, as demonstrated by the last column of Table 3. This contrasts with the common belief that parameter sharing primarily improves computational efficiency (Li et al. (2021)).

## 5.2 EVALUATION OF KAANS

**Single Layer Network**  To assess the fundamental representational capacity , we implement several single layer models to be test including SLP with ReLU, SiLU and Tanh, single layer KAN, and KAANs with different activations. In each training round, we use SciPy to create the Toy dataset, and all models are trained and tested under the same conditions for 100 epochs. In classification, blobs, circles and moons, the metric is accuracy, while in friedman1, it is Mean Squared Error(MSE).

The averages of all 100 rounds are shown in Table 4, where all the best results of each dataset belong to KAANs. Although KAANs with different activation functions perform well, the optimal activation function varies across different tasks. Therefore, KAAN, demonstrates strong representation capacity under single-layer configuration.

**Multi-Layer Network**  To further evaluate the representational power of KAAN, we constructed a deeper model. We modify the aforementioned single-layer structures into three-layer structures, including one hidden layers. In each round, we train these models on tabular benchmarks for 550 epochs, with the best test accuracy for each model recorded.

The averages and standard deviations of all 10 rounds are shown in Table 5, indicating that KAANs with different activations perform all well. KAAN with DoG does not obtain any optimum in Table 4, but here it achieve five optima. In the Poly16 model, gradient explosion leading to NaN loss often occurs within 400 epochs, resulting in training failure. This aligns with our claim in Section 2, that higher-order polynomials are not suitable activation functions in more complex neural networks.The

Table 4: Accuracies/MSE of MLPs, Single Layer KAN and Single Layer KAANs

| Model | classification(Acc) | blobs(Acc) | circles(Acc) | moons(Acc) | friedman1(MSE) |
|---|---|---|---|---|---|
| KAN | 98.58 | 83.67 | 99.00 | 94.81 | 19.21 |
| SLP_tanh | 98.85 | 88.23 | 49.46 | 71.13 | 9.79 |
| SLP_relu | 98.85 | 86.65 | 51.00 | 70.50 | 12.87 |
| SLP_silu | 98.84 | 86.28 | 49.44 | 85.81 | 10.38 |
| KAAN_PV1 | 99.02 | 89.53 | 89.27 | 89.44 | 3.10 |
| KAAN_PV2 | **99.08** | 90.06 | 99.12 | **99.89** | 1.94 |
| KAAN_F | 99.06 | **90.08** | 99.11 | 99.60 | **1.91** |
| KAAN_P4 | 99.06 | 89.58 | 98.42 | 91.34 | 1.99 |
| KAAN_P16 | 99.07 | 89.70 | 98.43 | 91.79 | 2.73 |
| KAAN_P4*4 | 99.08 | 89.94 | 99.01 | 97.56 | 2.02 |
| KAAN_G | 99.05 | 89.97 | **99.17** | 88.43 | 1.93 |
| KAAN_DoG | 99.07 | 90.01 | 99.17 | 99.63 | 1.93 |

Table 5: Accuracies of MLPs, Multi Layer KAN and Multi Layer KAANs

| Model | BCWD | Spambase | Dry Bean | Adult | MAGIC | Statlog | CDC | Poker Hand |
|---|---|---|---|---|---|---|---|---|
| KAN | 71.93 | 94.67 | / | 76.20 | 64.68 | 99.31 | 84.66 | 56.84 |
| MLP_tanh | 97.81 | 93.97 | 93.09 | 85.90 | 86.34 | 99.90 | 85.13 | 62.72 |
| MLP_relu | **98.68** | 94.02 | 93.22 | 85.94 | 86.28 | 99.85 | 85.00 | 55.11 |
| MLP_silu | 96.49 | 94.13 | 93.22 | 85.66 | 87.00 | 99.87 | 85.10 | 55.42 |
| KAAN_Pv1 | 97.37 | 94.46 | 93.37 | 86.14 | 88.56 | 99.96 | 84.98 | 60.63 |
| KAAN_PV2 | 97.37 | 95.05 | 93.44 | 86.11 | 88.41 | 99.96 | 85.03 | 58.39 |
| KAAN_F | 98.25 | 94.89 | 93.09 | 85.85 | 88.17 | 99.90 | 85.08 | **69.12** |
| KAAN_P4 | 96.05 | 94.78 | **93.96** | 86.20 | 87.67 | 99.84 | 85.08 | 62.09 |
| KAAN_P4*4 | 96.93 | 94.24 | 93.33 | 86.17 | 87.80 | 99.81 | 84.98 | 61.72 |
| KAAN_P16 | 98.68 | / | / | / | / | / | / | / |
| KAAN_G | 97.37 | 94.67 | 92.65 | 86.05 | 88.47 | 99.89 | 84.96 | 58.85 |
| KAAN_DoG | 97.37 | **95.54** | 93.31 | **86.31** | **88.76** | **99.97** | **85.17** | 63.04 |

The notation '/' denotes that this model fails in all rounds of training on this dataset.

performance of KAN in Table 4 and Table 5 also aligns with the statement that KAN exhibits weak performance in complex tasks (Le et al. (2024)).

**Convolutional KAAN**  In Section 4, we discuss the application of KAAN within contemporary neural network architectures, with a focus on its implementation in convolutional neural networks (CNNs). In Section 5.1, we validate four different CKAAN structures utilizing ReLU activation and show the results in the last four columns of Table 3. To further evaluate this methodology on more activations, we construct a CKAAN-enhanced ResNet50 by replacing the second convolutional layer and the corresponding activation layer with a CKAAN layer. We conduct experiments on the CIFAR100 dataset, ensuring that all experimental settings remained consistent with the baseline. The performance of CKAAN-PV1 and CKAAN-PV2 was then compared against the standard ResNet50.

The experiment follows a two-stage training process. The first training phase of the model is a 10-epoch warming-up with learning rate $1e-5$. Then we train models for 90 epochs using learning rate $1e-4$.

Table 6: Accuracies of CKAANs and Standard ResNet50

| | Standard ResNet50 | CKAAN_PV1 | CKAAN_PV2 |
|---|---|---|---|
| **CIFAR100** | 66.80 | **69.62** | 59.02 |

As shown in Table 6, CKAAN achieved the best performance. It is also evident that ParallelV1, which uses fewer activation components, significantly outperforms ParallelV2 in this experimental setup. This could be due to the excessive fitting components in ParallelV2 or the potential conflicts between some of these components.

# 6 Discussion and Conclusion

By delving into the similarity of KAN and MLP, we propose KAAN where activation is a linear combination of any univariate and continuous basis components. From a structural perspective, our approach is evidently morestraightforward, flexible, and easy to apply to modern well-established networks than KAN with B-splines. In terms of straightforwardness, the B-spline method relies on control points for adjustments, but changes to each point often affect two grids, making local adjustments complex and difficult to understand. In contrast, our method employs a linear combination of multiple independent fitting components. This allows us to clearly understand the contribution of each component and straightforward control the overall outcome. Regarding flexibility, B-splines rely on pre-set control points and generate curves through recursive calculations. Our approach consists of multiple independent components, allowing us to adjust each component individually without affecting the overall structure. Additionally, we can flexibly add or prune components as needed, enabling precise control over both local and global structure, which significantly enhances the adaptability and scalability of the model. With respect to transferability, KAAN essentially only swaps the execution order of linear transformation and activation function of the network, yet this achieves the effect of splitting a neuron's edge into multiple parts. For all methods that use neurons, KAAN can directly replace the edges in their neural networks.

As we discussed in Section 5, the optimal activation varies in different tasks. We have only tested a few of the commonly used basis functions that have been employed in fitting tasks and LANs. Although these basis functions are relatively representative, the proportion we tested is insignificant compared to the vast group of basis functions used in fitting tasks. Therefore, we believe that researchers still need to further explore the possibilities of basis functions.

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
