# OpenReview forum: "KAAN: Kolmogorov-Arnold Activation Network --- a Flexible Activation Enhanced KAN"
_ICLR.cc/2025/Conference — Submitted to ICLR 2025_

### Official Review · Reviewer_nBC1 · 2024-10-16

**Soundness:** 3
**Presentation:** 3
**Contribution:** 3
**Rating:** 6
**Confidence:** 3

**Summary:**

The authors proposed a novel framework of viewing MLPs and a special case of KANs and proposed as a inspiration KAAN, where each nonlinear activation function is parametrized by a linear combination of basis functions. They conducted extensive experiments on challenging datasets including Tabular datasets and Cifar-10, and introduced a convolutional version as well. The article presented an interesting perspective and should be treated as a nice improvement on KANs, with the following limitations.

1. While KAAN seems interesting, it seems still such a way of parametrization of nonlinearity in KANs, with more complicated nonlinearity. This improvement is at best incremental and would need more support from numerical evidence.

2. The referee would envision that KAANs suffer from less interpretability than KANs; especially on symbolic regression. Could the authors comment on this restriction?

3. It would be interesting to elaborate more on the perspective in Sec 3.2 and gain more motivation on the comparison between KANs and MLPs.

4. How does (C)KAAN perform on more challenging tests?

**Strengths:**

1. The authors proposed a novel framework of viewing MLPs and a special case of KANs

2. They conducted extensive experiments on challenging datasets including Tabular datasets and Cifar-10, and introduced a convolutional version as well.

**Weaknesses:**

1. While KAAN seems interesting, it seems still such a way of parametrization of nonlinearity in KANs, with more complicated nonlinearity. This improvement is at best incremental and would need more support from numerical evidence.

2. The referee would envision that KAANs suffer from less interpretability than KANs; especially on symbolic regression. Could the authors comment on this restriction?

**Questions:**

1. It would be interesting to elaborate more on the perspective in Sec 3.2 and gain more motivation on the comparison between KANs and MLPs.

2. How does (C)KAAN perform on more challenging tests?

---

> ### Author Response · Authors · 2024-11-27
>
> We sincerely thank the reviewer for the thorough and constructive feedback and suggestions, and we appreciate the support for this manuscript.
>
> # Response to the concern in Weakness 1 about interpretability on symbolic regression.
>
> Regarding interpretability in symbolic regression, we identify two key criteria:
>
> 1. **Traceability**: Whether the relationship between model inputs and outputs can be easily traced.
> 2. **Physical Accuracy**: Whether the model can accurately discover formulas aligned with physical principles.
>
> For the first criterion, we acknowledge that B-spline offers good interpretability once the curve is generated. However, the recursive computations required to relate its parameters to generated curves significantly increase the difficulty of human cognitive understanding. Additionally, in a three-degree spline grid with second-degree smoothing that ostensibly has four parameters, only one parameter is explicitly available, with the remaining three determined by smoothing constraints—a framework that is not intuitive for human cognition.
>
> For the second criterion, B-spline struggles to achieve this independently and often requires human assistance. In contrast, our activation function composition is fundamentally a gradient-based symbolic regression method. We can derive concise functional expressions by reconstructing the target function and applying KAN’s pruning methods at the component level within KAAN.
>
> # Response to concern in Question 2 about manuscript organization
>
> We have revised our manuscript to include an explanation of the conditions for this comparison, aiming to provide more context and motivation from the comparison between KANs and MLPs, and construct KAAN using a more concise approach.
>
> We hope our responses can address the reviewer’s concerns.

---

> > ### Comment · Reviewer_nBC1 · 2024-11-27
> >
> > Thanks; i will thus keep my score

---

### Official Review · Reviewer_YTfN · 2024-10-29

**Soundness:** 3
**Presentation:** 3
**Contribution:** 2
**Rating:** 5
**Confidence:** 4

**Summary:**

This paper introduces a novel architecture named KAANs, which enhances the efficiency of MLP by incorporating a method inspired by KANs. Theoretically, the paper begins by establishing that MLPs are a subset of KANs and then deviates from traditional KANs by replacing B-spline activation functions with linear combinations of basis functions. Experimentally, the paper evaluates 7 different combinations of basis functions as activation functions across various AI-related tasks, demonstrating that KAANs achieve higher accuracy than both MLPs and KANs.

While the theoretical foundation is robust and compelling, the KAANs just replace activation functions in MLPs with more complex functions. when trying to search for the optimal combination of basis functions along with the most effective weights, the concept go back to the learnable activation functions.  Therefore, it appears that the paper has elegant theory but not enough contributions on practical level.

**Strengths:**

The theoretical framework is elegantly and solidly constructed.

It points out that “MLP represents a specific instance of KAN”

It points out that“any continuous univariate basis functions can be used as activation function”

KAANs offer greater flexibility and fewer limitations than traditional KANs, making them more adaptable to various structures.

The paper conducts extensive experiments across a multitude of AI-related tasks.

**Weaknesses:**

The paper experiments with various combinations of basis functions, where different combinations excel in different tasks. This variability raises questions about how to determine the most effective combination for a given task.

Although KAANs outperform MLPs and KANs in the experiments, the comparison may not be entirely fair. The more complex activation functions used in KAANs require greater computational power compared to MLPs, potentially skewing the results. Similarly, comparing KAANs to KANs without adjusting for KANs' longer training requirements may not provide a balanced view of their respective efficiencies.

**Questions:**

My opinion could shift towards acceptance if the authors could address one of the following points:

Develop a method to identify the most optimal combination of basis functions.

Find a specific combination of basis functions that consistently outperforms others.

Demonstrate that in certain specific tasks, KAANs offer a significant advantage.

---

> ### Author Response · Authors · 2024-11-27
>
> We sincerely thank the reviewer for the thorough and constructive feedback and suggestions. Our responses to each point are as follows:
>
> >Develop a method to identify the most optimal combination of basis functions.
>
> report (Liu et al. 2024) has already proposed a solution that involves batch or manual pruning. This method statistically analyzes the input and output scales for each edge during training and uses their scale ratio as a measure of the edge's significance for pruning. While effective in KAN, this method is computationally expensive. Although we have not tested it ourselves, we believe that a similar statistical analysis could be applied to our method to identify the optimal configuration for each edge, yielding results comparable to those in KAN. This also aligns with our expectation of customizing neural networks with varying properties and structures for each edge. However, since this method is proposed by the report (Liu et al. 2024), we did not include it in our paper. We sincerely apologize for any inconvenience caused by this omission.
>
> >Find a specific combination of basis functions that consistently outperforms others.
>
> It is unlikely to find a single combination of activation functions that is universally optimal in all cases.
>
> In recent years, many well-known networks have demonstrated performance improvements by changing activation functions from ReLU to GeLU, as well as contrary cases where GeLU was replaced with ReLU in ConvNext v2. Other activation function variants like Leaky ReLU and Swish have also been used. Across different tasks, modifying activation functions has often led to performance enhancements and the emergence of numerous state-of-the-art results. This variability is perplexing, and it raises the question of whether these changes are meaningful or arbitrary.
>
> We have observed that recent work, such as (Poeta et al. (2024)), examined tabular benchmarks with similar data formats but significant differences in the domains of data sources. By selecting basis functions with substantial differences, it is possible to explore whether certain functions consistently outperform others in datasets from different domains with such large variations. Following a similar rationale, we chose large-span toy datasets when selecting activation functions for single-layer KAAN. Our results suggest that the dominant activation function depends on the task and structure, confirming the variability in this problem.
>
> Therefore, although we also hoped to identify a universally optimal combination, we must respect the experimental conclusion, which strongly indicates the opposite.
>
> >Demonstrate that in certain specific tasks, KAANs offer a significant advantage.
>
> As shown in Table 4 and Table 5 of our manuscript, KAN and MLP often have their respective strengths and weaknesses. Sometimes, one or both of them perform poorly on certain problems such as Friedman1, Circles, Moons, BCWD, Adult, and MAGIC. However, our method does not exhibit such issues.
>
> Once again, we sincerely appreciate the reviewer for the thorough and constructive feedback and suggestions. We hope our responses can address the reviewer’s concerns.

---

### Official Review · Reviewer_Htn9 · 2024-11-01

**Soundness:** 3
**Presentation:** 2
**Contribution:** 2
**Rating:** 3
**Confidence:** 3

**Summary:**

This paper presents Kolmogorov-Arnold Activation Networks, an extension of Kolmogorov-Arnold Networks, that uses MLP/CNN-like architecture with flexible activation functions defined for each edge between neurons.

**Strengths:**

1. The proposed approach clearly works on the presented tasks, and in some cases provides a performance improvement.
2. The KAAN parametrization is compatible with standard ANN architectures.
3. Related to the previous point, this parametrization might be helpful for neural architecture search/meta-learning/similar approaches that adapt neural networks’ architectures, as the nonlinearity parameters are designed to be differentiable.

**Weaknesses:**

2. Memory and computation time requirements

The computational requirements of KAANs appear to be much higher than for corresponding standard MLPs/CNNs. Eq. 6 uses several weights per connection (one for each activation type) and additionally parametrizes the activations. This should increase both memory consumption and running time of KAANs compared to standard networks.

The increased number of parameters in KAANs also (unless I missed something) suggests the performance improvements (Tabs. 3-5) are very modest compared to standard networks that use several times fewer parameters.

3. Lack of interpretability

Throughout the paper, KAANs are called intuitive. However, I do understand how KAANs are more intuitive than standard MLPs (if anything, they are more convoluted). The results in Tabs. 3-5 indirectly confirm my concern: there’s no clear winner across different combinations of activation functions.

Lines 300-311 discuss the potential uses cases for each activation function, but all of those apply to standard ANN architectures that don’t define edge-based nonlinearities.

4. Poor writing

The paper needs some writing improvements. Here are some instances I’ve noticed, although text needs overall polish.
1. [Line 30] “There were not many breakthroughs until KANs” [rephrased] – I would disagree, and suggest Transformers as an obvious architectural breakthrough. But, the list can expand with for instance capsule networks (https://www.sciencedirect.com/science/article/pii/S1319157819309322) and gflownets (https://arxiv.org/abs/2111.09266).
2. The introduction contains many terms, such as LANs and TANs, but they’re not cited until related work.
3. “No many” instead of “not many” in line 30, extra bracket in line 81, typo in line 205, non-plural “Experiment” name for Sec. 5

**Questions:**

1. What are the parameter counts/VRAM consumption/running time for the tested KAANs vs. MLPs/CNNs?
2. Is it possible to compare KAANs with standard networks that use the same number of parameters?

---

> ### Author Response · Authors · 2024-11-27
>
> We sincerely thank the reviewer for the thorough and constructive feedback and suggestions. We hope our responses can address the reviewer’s concerns.
>
> # Response to concerns in Weakness 2 & 3 and Question 1 & 2 about parameter counts
>
> In our study, while we primarily compare our work to models derived from MLP-based frameworks, the foundation of our approach actually stems from KAN. Our goal is to preserve the strengths of KAN’s EDGE-centric paradigm while leveraging the engineering simplicity offered by MLP. As such, we use KAN as the reference point for evaluation, both in terms of conceptual clarity and parameter performance. These weaknesses stem from issues commonly faced by the KAN family: it tends to have higher computational costs and parameter counts. This aligns with one of KAN’s original design goals—to pack more parameters into the same neural topology, addressing the scalability bottlenecks typically encountered by MLP.
>
> # Response to concerns in Weakness 3 about the absence of a champion.
>
> In recent years, we’ve observed that many leading networks gain performance improvements by switching activation functions, such as replacing ReLU with GeLU, or even reverting back as in the case of ConvNext v2. Other studies have also explored variants like Leaky ReLU and Swish, with some achieving notable gains and producing numerous state-of-the-art results. This frequent switching of activation functions across architectures has generated considerable curiosity about the actual significance of these changes.
>
> Although activation functions are not the primary focus of our research, the problem we investigate provides an opportunity to test whether a universally superior activation function exists across datasets that share similar structures but vary in intrinsic characteristics. Inspired by (Poeta et al.’s (2024)) tabular benchmarks, which present structurally alike datasets but originate from diverse domains, we conducted experiments using a range of activation functions. We explored the question rigorously by deploying these functions and their combinations within the KAAN model. Our results confirmed that no single activation function consistently outperforms others across all scenarios, reaffirming the issue's complexity.
>
> # Response to concerns in Weakness 3 about applications of activation functions
>
> We sincerely apologize for causing such a misunderstanding. In lines 300 to 311, we delve into the variety and origins of activation functions, explaining how each has proven effective in specific types of problems. By illustrating their diverse applications, we highlight the significant differences in their properties while emphasizing their exceptional performance in their respective domains.
>
> # Response to concerns in Weakness 4 about manuscript writing
>
> We have revised our manuscript and modified some expressions to enhance the reading experience of the article.
>
> Once again, we sincerely appreciate the reviewer for the constructive comments.

---

> > ### Comment · Reviewer_Htn9 · 2024-11-27
> > **Response to rebuttal; keeping the same score**
> >
> > Thank you for the response! I appreciate the writing fixes, but otherwise I consider the weaknesses and questions I raised unresolved: the computational costs of KAANs compared to MLPs are indeed higher without significant performance gains (it is also still not clear what the actual parameter count for the models is) and the interpretability benefits are not clear from the results. Therefore I will keep the same score.

---

### Official Review · Reviewer_FmRt · 2024-11-05

**Soundness:** 2
**Presentation:** 3
**Contribution:** 1
**Rating:** 3
**Confidence:** 3

**Summary:**

Authors propose Kolmogorov-Arnold activations inspired from KANs (Kolmogorov-Arnold Networks) and replace B-splines in KANs to achieve similar or better performance than MLPs. Authors show that MLPs can be represented in a form conforming to Kolmogorov-Arnold representation Theorem (KAT). Using MLP-like equipped with Kolmogorov-Arnold activations, authors experiment and compare different basis functions. Experiments also demonstrate successful integration with Convolutional Neural Networks (CNNs) which achieving comparable performance.

**Strengths:**

1. The paper is written clearly and concisely, and is easy to read.
2. The proposed activation makes KANs more flexible and easy to deploy which would encourage the scientific community to experiment with these networks.
3. Experiments clearly demonstrate that the proposed activation function allows KANs to be trained while achieving comparable performance to MLPs and even ResNets.

**Weaknesses:**

1. Novelty is missing: KAN arxiv report (Liu et. al 2024) already gives a MLP-like interpretation of KANs which allows stacking of layers similar to MLPs which is similar to section 3 in the paper.
2. Authors have essentially replaced splines, which is a core contribution of the original KAN paper (provides higher degree of control to model univariate functions) with learnable activation functions. There is already literature covering learnable activation functions with different basis like Polynomial or sinusoidal basis (in context of MLPs). Therefore I feel the paper doesn’t bring new insights into Neural Networks or KANs.

**Questions:**

1. I would suggest authors to reevaluate the core contributions and rewrite the paper. If the main contribution is empirical in nature, I would suggest doing more experiments on transformer-like architectures or showing taks where MLPs or KANs fail to learn underlying functions correctly but the proposed method can.
2. What is the meaning of “KAN faces the challenges of being unintuitive and inflexible.” This is a highly subjective statement, giving concrete examples of what inflexible and unintuitive means would help readers. Does KAAN help give more flexibility or intuition? If so, how? What is the takeaway?

---

> ### Author Response · Authors · 2024-11-27
>
> We sincerely thank the reviewer for the thorough and constructive feedback and suggestions. We hope our response can address the reviewer’s concerns.
>
> # Response to concern in Weaknesses 1 about novelty.
>
> The content described in Chapter 3 can be summarized as follows: Each layer in an MLP involves two operations. By taking one operation from the previous layer and another from the next layer of the MLP, a new layer is formed that satisfies the KAT requirement. (Liu et al. (2024))  believes that they are different. Please allow us to quote the original text as follows:
>
> >It is clear that MLPs treat linear transformations and nonlinearities separately as $\mathbf{W}$ and $\sigma$, while KANs treat them all together in $\mathbf{\Phi}$. In Figure 0.1(c) and (d), we visualize a three-layer MLP and a three-layer KAN, to clarify their differences.
>
> We speculate that the reviewer might be referring to the description in Section 2.2 about stacking KAN. At the beginning of Section 2.2, the authors note that MLP stacking relies on its layer structure, and KAN can also construct layers similarly to MLP, which allows it to form a deep network. While they use the term "analogy," they clearly do not imply that KAN is an MLP-like structure. Importantly, we have proven that MLP is a subset of KAN, not the other way around.
>
> The parameters of the B-spline activation function chosen in KAN are non-separable, so a B-spline-based KAN cannot be represented as an MLP structure. Our proof relies on the associativity and distributivity within the local structure of activation-affine-activation, which KAN lacks. If the reviewer refers to the description at the beginning of Section 2.2 about stacking KAN, we believe our Chapter 3 content is entirely different. Please let us know if you are referring to another part of the paper or using a version other than 2404.19756v4. Thank you very much.
>
> # Response to concerns in Weakness 2 and Question 1 about contributions
>
> To better showcase our work, we summarize all the contributions of our paper as follows:
>
> 1. **Proof Establishment**: We first provided the aforementioned proof.
> 2. **KAAN Network**: Based on these findings, we proposed rearranging the order of linear transformations and activation functions, which facilitates the construction of edge-centric networks. As all connections in the network are various activation functions, we named it KAAN.
> 3. **Ease to Be Transfered**: Our architecture makes constructing KAT-compliant networks easy. Rearranging the order of linear transformations and activation functions in mature structures only requires this approach, which is easily extendable to convolutional domains.
> 4. **Experimental Validation**: We demonstrated that rearranging the order of linear transformations and activation functions does not degrade the performance of well-designed network architectures when shifting from the UAT paradigm to the KAT paradigm.
> 5. **Independent Activation Learning**: KAN enables independent learning for each activation function. However, modern structures involve extensive neuron reuse, and we tested whether such reuse should be retained.
> 6. **New Insights on Convolution**: We found that reusing convolution kernels not only improves efficiency but also that providing independent convolution kernels for each output pixel severely impacts performance, challenging the conventional understanding of convolution kernel reuse.
> 7. **Superior Performance on Tabular Data**: KAAN achieved optimal results across multiple benchmarks on tabular datasets with different backgrounds due to its ability to choose fitting bases from a broader range.
> 8. **Task-Specific Activation Functions**: We discovered that the dominant activation function varies across datasets with different features. Even for similar data types and tasks, datasets with different content may favor different types of activation functions.
>
> The primary contributions of this paper are outlined as follows:
> - Points 1-3 are the core contributions.
> - Points 4-7 provide performance validations for contributions 2 and 3.
> - Points 6 and 8 represent new insights gained during the proof process.
>
> This paper does not aim to identify a globally superior activation function, and our experiments do not support the existence of such a function.
>
> # Response to concerns in Question 2 about manuscript writing
>
> We revised our manuscript, correcting parts that might cause misunderstandings and replacing them with more complete and detailed descriptions.

---

> > ### Comment · Reviewer_FmRt · 2024-12-03
> > **Response to authors**
> >
> > I thank authors for their response. I understand the contributions as highlighted in the paper, but the important questions is what is the utility of KANs or similar networks? From the original KAN report (Liu et. al 2024), these networks perform better on small tasks with compositional nature and can also provide higher interpretability. The main motivation to improve upon these networks in not fully clear, I understand KAN works well on certain tasks and authors improve their applicability and transferability. But does the proposed design work better than MLPs with learnable activation functions? I think this question needs to be answered in the introduction and discussed in experiments. Therefore I am keeping my rating the same.

---

### Meta-Review · Area_Chair_UEA5 · 2024-12-15

**Metareview:**

The paper describes a variant of the so-called "Kolmogorov-Arnold Network" (KAN). The authors note the similarity of the KAN to a classical MLP, and they propose a variant where, instead of B-splines, they use a small one-input-one-output MLP as edge activation function.

The paper had a very negative round of reviews, with multiple reviewers recommending a definite reject. They lamented a lack of novelty, marginal improvements in the experiments, insufficient related works (e.g., on trainable activation functions), and poor writing. Rebuttal came very late and did not address the issues.

In general, I agree with the reviewers' considerations. I see no reason to overrule their consensus, and I recommend a rejection of the paper.

**Additional Comments On Reviewer Discussion:**

- **Reviewer nBC1** was concerned about poor interpretability and marginal improvements in the results. Rebuttal was not convincing.

- **Reviewer YTfN** was concerned about variable results across datasets, and the increased computational complexity of the method. While they considered a potential change in score, the rebuttal came very late and there was no further discussion.

- **Reviewer Htn9**, similarly to the previous two, was concerned about interpretability of the method and computational complexity. He also lamented poor writing across the paper. Rebuttal was not convincing.

- **Reviewer FmRt** highlighted that the MLP interpretation of KANs is known (even from the original paper) and that the paper is lacking a serious related work section on trainable activation functions, which are a vast subfield of neural networks.

In general, all reviewers were negative. *I agree with most of their points* and they weighted equally in my evaluation.

---

### Decision · Program_Chairs · 2025-01-22

Reject